# Optimization of Antioxidant Hydrolysate Produced from Tibetan Egg White with Papain and Its Application in Yak Milk Yogurt

**DOI:** 10.3390/molecules25010109

**Published:** 2019-12-27

**Authors:** Fumin Chi, Ting Liu, Liu Liu, Zhankun Tan, Xuedong Gu, Lin Yang, Zhang Luo

**Affiliations:** 1College of Food Science, Tibet Agriculture & Animal Husbandry University, Nyingchi 860000, China; tanzhankun@foxmail.com (Z.T.); guxuedong78@163.com (X.G.); yanglintibet@sina.com (L.Y.); luozhang1759@sohu.com (Z.L.); 2College of Food Engineering and Nutrition Science, Shaanxi Normal University, Xi’an 710119, China; liuting-1430@snnu.edu.cn

**Keywords:** Tibetan egg white, papain hydrolysis, antioxidant activity, yak milk yogurt

## Abstract

The objective of the present study was to produce antioxidant hydrolysate from Tibetan egg white protein hydrolyzed with papain, and to investigate the effect of added papain egg white hydrolysate (PEWH) on the quality characteristics and amino acid profiles of yak milk yogurt. A response surface methodology (RSM) was utilized to analyze the effects of hydrolysis time (X_1_), the ratio of enzymes to substrates, and enzyme dosage (X_3_) on the superoxide anion radical (O_2_^−^) scavenging activity of hydrolysates. The predicted maximum value of O_2_^−^ scavenging activity (89.06%) was obtained an X_1_ of 2.51 h, X_2_ of 4.13%, and X_3_ of 4500 U/g of substrate, almost approaching the experimental value (88.05 ± 1.2%). Furthermore, it was found that the addition of PEWH to yak milk can enhance acidification, sensory score, the number of lactic acid bacteria (LAB), and the amino acid content in yak milk yogurt. The results suggested that PEWH displayed an exceptional potential to be developed as a functional food ingredient that could be applied during the manufacturing process of yak milk yogurt.

## 1. Introduction

As a vital source of macro and micronutrients within the diet, eggs are comprised of proteins, lipids, vitamins, and minerals and are one of the most prevalent foods across the globe [1]. The Qinghai-Tibet Plateau is the highest in the world. Tibetan chickens are considered an excellent local poultry breed and inhabit altitudes of 2,900 m, while their eggs have a hatchability of approximately 88.9% [2]. Jia et al. [3] demonstrated that the adaptation of Tibetan chicken to high altitudes might be associated with higher quantities of yolk in the egg and low metabolic oxygen demand in tissue. Tibetan eggs possess significant genetic diversity, providing them with outstanding adaptability and food quality. Chen et al. [4] reported that the iron, zinc, and calcium content in Tibetan eggs is higher than in regular eggs.

The antioxidant peptides and hydrolysates extracted from food proteins can be regarded as natural, safe, and beneficial food ingredients exhibiting low molecular weight, low cost, high biological activity, and easy absorption [5,6]. Recently, bioactive peptides derived from egg proteins have been reported to fulfill various biological functions, such as antioxidant, antimicrobial, antitumor, anti-inflammatory, antihypertensive, and antidiabetic activity [7,8]. Egg white protein provides essential nutrients for embryonic development and is a source of peptides (VYLPR, EVYLPR, VEVYLPR, and VVEVYLPR) with high antioxidant activity [9,10]. Liu et al. [11] isolated and identified three novel antioxidant peptides (DHTKE, FFGFN, and MPDAHL) from the hydrolysates of egg white protein, which may have application potential as functional food ingredients. Using the iTRAQ technique, Wu et al. [12] revealed that the most abundant proteins in the Tibetan ovalbumin peptide included vitellogenin-2, vitellogenin-1, vitellogenin, apolipoprotein A-I, and serum albumin.

Bioactive peptides released from various egg proteins with enzymatic proteolysis, act as potential physiological modulators of metabolism. Proteases, such as papain, trypsin, pancreatin, and pepsin, are typically used to prepare hydrolysates and peptides [7]. Garces-Rimon et al. [13] revealed that pepsin egg white hydrolysate improves obesity-related oxidative stress, inflammation, and steatosis in Zucker Fatty Rats. Moreno-Fernandez et al. [14] reported that the oral administration of pepsin egg white hydrolysate could be used as a functional ingredient to improve metabolic syndrome. Lorenzo et al. [5] indicated that future studies associated with natural antioxidant peptides should also explore the protective strategies of antioxidant peptides in food matrices.

Fermented yak milk products are the most commonly consumed dairy foods around the Qinghai-Tibet Plateau while possessing a unique yak milk flavor and are rich in nutrients [15]. Recent years have seen increasing interest in applying natural functional additives to dairy products [16], and adding bioactive antioxidant hydrolysates and peptides to dairy products can enhance the viability of probiotics [17]. Therefore, the antioxidant potential of PEWH is evaluated. Hydrolysis conditions are optimized using RSM, to facilitate the harvest PEWH exhibiting the highest O_2_^−^ scavenging activity. In addition, further investigation is conducted regarding the effect of added PEWH on the physicochemical properties, sensory characteristics, and amino acid content of yak milk yogurt during the fermentation process.

## 2. Results and Discussion

### 2.1. The Prediction Capability of the Regression Model

The optimization of RSM is a useful technique for investigating complex process conditions by establishing a mathematical model and is superior to the traditional single parameter and orthogonal-test optimization [18]. The methodology has been widely applied to optimize the hydrolysis conditions of bioactive hydrolysates from some animal proteins. In addition, the RSM is employed to investigate the interactions between independent variables by establishing a model. Therefore, the method for statistical data analysis/RSM was an efficient statistical tool in the optimization of the hydrolysis conditions [19,20,21].

In this study, a BBD of RSM was employed to study the effects of three variables (X_1_, X_2_, and X_3_) on the O_2_^−^ scavenging activity of PEWH. Three independent variables and their variation ranges were selected based on the results of the preliminary experiment and other related reports [19,20,22]. The operational conditions and experimental results for the responses are shown in Table 1. Furthermore, the quadratic polynomial regression equation was obtained to express the O_2_^−^ scavenging activity of PEWH as a function of the independent variables in this work. The following equation can describe the model:Y = −561.85669 − 20.18150X_1_ + 266.28050X_2_ + 0.060686X_3_ + 12.39500X_1_X_2_ + 2.88500 × 10^3^X_1_X_3_ − 9.68250 × 10^3^X_2_X_3_ − 8.75975X_1_^2^ − 30.74725X_2_^2^ − 3.59225 × 10^6^X_3_^2^(1)
where Y is the predicted O_2_^−^ scavenging activity of PEWH, and the X_1_, X_2_, and X_3_ represent hydrolysis time, substrate mass fraction, and enzyme dosage, respectively. Further investigation occurred regarding the significance of the terms of the mathematical regression model based on the variance analysis of BBD. The *F*-value and *P*-value were employed to express the significance between each coefficient, and the results are shown in Table 2. The corresponding variables would be more significant if the absolute *F* value becomes greater, and the *p*-value becomes smaller [23]. As shown in Table 2, three linear coefficients (X_1_, X_2_, and X_3_), two cross-product coefficients (X_1_X_2_, and X_2_X_3_), and two quadratic term coefficients (X_1_^2^ and X_2_^2^) were significant (*P* < 0.05), whereas the other coefficients (X_1_X_3_ and X_3_^2^), were not significant (*P* > 0.05). Furthermore, the *P*-value of the model was extremely small (*P* < 0.001), which means that the model displays significant. The *P*-value for the lack of fit displays insignificant (*P* = 0.1273), confirming the validity of the model. These results suggested that the model had good fitting accuracy and could be used for the optimization design.

R^2^ is the coefficient of determination that is defined as the ratio of the explained variation to the total variation. When the value of R^2^ approaches unity, the empirical model fits the actual data [24]. As shown in Table 2, the value of R^2^ was 0.9803, indicating that the model did not explain 1.97% of the total variations. The results suggested that the model exhibited good correlation and was suitable for predicting the actual values within the range of the tested variables. Moreover, the value of the adjusted R^2^ was 0.9549 and the value of the C.V.% was 7.36, also confirming minimal variability and the high stability of the model.

### 2.2. Response Optimization

Three-dimensional (3D) and two-dimensional (2D) contour plots were constructed to illustrate the mathematical regression model, and the results are presented in Figure 1. Three independent variables (X_1_, X_2_, and X_3_) were obtained by keeping one of the variables constant at the center point while varying the other two. The plots indicated the effect of hydrolysis conditions on the O_2_^−^ scavenging activity of PEWH.

The effect of X_1_ and X_2_ on the O_2_^−^ scavenging activity of the hydrolysates was illustrated in the 3D and 2D contour graphs (Figure 1A,B), indicating that the antioxidant activity of PEWH was affected by X_1_ and X_2_, while X_3_ remained constant at 5500 U/g. The O_2_^−^ scavenging activity of the hydrolysates increased when extending X_1_ from 2 h to 3 h, and X_2_ ranged between 3.0% and 3.9%, after which it reached a peak point. However, the O_2_^−^ scavenging activity of the hydrolysates decreased when X_1_ exceeded 3 h and X_2_ exceeded 4.2%.

The effect of X_1_ and X_3_ is illustrated in the 3D and 2D contour graphs (Figure 1C,D), indicating that the O_2_^−^ scavenging activity of the hydrolysates was affected by X_1_ and X_3_, while X_2_ remained constant at 4% of the center value. The O_2_^−^ scavenging activity decreased when X_3_ was increased from 4500 U/g to 6500 U/g. Furthermore, the O_2_^−^ scavenging activity gradually increased when X_1_ was extended from 2 h to 2.4 h, while a declining trend was observed with an extension from 2.4 h to 4 h.

The effect of X_2_ and X_3_ is illustrated in the 3D and 2D contour graphs (Figure 1E,F), indicating that the O_2_^−^ scavenging activity of the hydrolysates was affected by X_2_ and X_3_, while X_1_ remained constant at 3 h of the center value. The O_2_^−^ scavenging activity increased when X_2_ was enhanced from 3% to 4.2%, but further enhancement from 4.4% to 5.0% led to a decrease in scavenging activity. When the range of X_2_ was between 3.8 and 4.8, the O_2_^−^ scavenging activity of the hydrolysates decreased in conjunction with increased X_3_.

The predicted and actual optimum papain enzymatic hydrolysis conditions are shown in Table 3. The predicted O_2_^−^ scavenging activity of PEWH was 89.06% when the enzymatic conditions were optimal. The optimum conditions were further investigated and repeated three times to confirm the validity of the analyzed mathematical model. As shown in Table 3, the experimental value of the O_2_^−^ scavenging activity of PEWH was 88.05 ± 1.20%, which corresponded with the predicted value. The results demonstrated that the model was highly accurate and that enzymatic conditions could effectively predict the O_2_^−^ scavenging activity of PEWH. By using a BBD of RSM, Fang et al. [18] demonstrated that the maximum DPPH radical scavenging activity of the hydrolysates derived from the papain mediated reaction was 74.40%, and also suggested that the model displayed an excellent fit between the RSM model and the actual experimental data. Yu et al. [22] established a model to investigate the effects of X_1_, X_2_, X_3_, hydrolysis temperature, and pH value on the preparation of peanut polypeptide with an alcalase and found that peanut polypeptide displayed effective antioxidant and antibacterial activity while decreasing blood pressure.

### 2.3. Quality Characteristics of Yak Milk Yogurt

Changes in the pH and TA of the yak milk yogurt samples during the fermentation process are shown in Figure 2A,B, respectively. At 0 h and 3 h of yak milk yogurt fermentation (Figure 2A), the initial pH of the yogurt samples containing 1%, 2%, and 3% PEWH was significantly higher than that of the YP0 sample (no added PEWH) (*p* < 0.05). The fermentation of the yogurt samples continued, and after 6 h, no significant differences were evident between the pH values of the three test samples (YP1, YP2, YP3) and the control sample (*p* > 0.05). As shown in Figure 2B, the initial acidity was significantly lower in samples containing 1,2% and 3% PEWH compared with the control. When samples were fermented with added PEWH (1%, 2%, and 3%) for 6 h and 12 h, respectively, the TA of the yak yogurt significantly increased (*p* < 0.05). Dave and Shah [25] reported that the ingredients added to dairy products considerably affected the incubation time required to reach a pH of 4.5. Moreover, high TA values result from lactose fermentation by yogurt LAB, while the ongoing metabolic and enzymatic activity of the mixed cultures can increase the TA and decrease the pH of the yogurt during low-temperature storage [26].

### 2.4. The Effect of Added PEWH on the Amino Acid Content of Yak Milk Yogurt

A heatmap is a two-dimensional graphical matrix that displays individual values via different colored cells. A similar color in a particular cell signifies similar content of a certain substance. Furthermore, the heatmap can directly exhibit the high-content substances and represents a classification of all the tested samples using cluster analysis [27]. Therefore, the effect of added PEWH on the composition and content of amino acids during yak milk yogurt fermentation was further investigated, and the results are presented in the form of a heatmap in Figure 3.

Sixteen amino acids were detected in different yogurt samples, of which seven were essential amino acids. The heat map of the 16 amino acids is shown in Figure 3 and indicates that Glu and Pro denoted the highest amino acid content in the yogurt. Although essential amino acids are obtained from food, it is vital to be consumed in the proper proportion [28]. This study showed that during the different stages of yak milk yogurt fermentation, the content of each amino acid including essential amino acids in the samples with the added PEWH to the yak milk, was higher than in the yak milk without PEWH. Previous research indicated that amino acid profiles and peptide sequences play a crucial role in some biological activities [29].

### 2.5. The Sensory Scores of Yak Milk Yogurt

Figure 4 shows the sensory scores of the yak milk yogurt [30] in terms of appearance, texture, flavor, and acceptability and, therefore, the mean evaluation scores of these four factors fell within the acceptable range. The acceptability of YP0, YP1, and YP3 exhibited no significant differences (*p* > 0.05), while the acceptability of YP2 was significantly higher than the other three samples (*p* < 0.05). Furthermore, no significant differences were apparent in the texture scores between the four test samples (*p* > 0.05). The flavor value of YP2 was significantly higher than the YP0 and YP3 yogurts (*p* < 0.05), and the acceptability of YP2 was also significantly higher than in the other three yogurts (*p* < 0.05). These results suggested that adding PWEH to yak milk will have a positive impact on the sensory sensation of the yogurt and adding the appropriate amount of PEWH can even improve the sensory properties of yak milk yogurt. A similar study by Wang et al. [29] found that no differences were evident between the control and the yogurt containing <0.3% silkworm pupae peptide concerning the flavor, appearance, as well as body and texture.

### 2.6. Viable Counts of LAB in Yak Milk Yogurt

The viable counts of LAB in the yogurt were expressed by the sum counts of LAB. As shown in Figure 5, at the end of yak milk yogurt fermentation, the addition of different amounts of PEWH significantly increased the total viable counts of LAB in comparison with the control. This result indicated that the addition of PEWH promoted the growth and proliferation of LAB in yak milk and may also reduce the fermentation time of yak milk yogurt. The main reason for this is that milk is mainly composed of macromolecular proteins and contains only a small amount of non-protein amino nitrogen. The non-protein amino nitrogen (peptides and amino acids) can enhance the growth and acid production capacity of *Streptococcus thermophilus*. In this study, the addition of PEWH to yak milk provides enough peptides and amino acids for the fermentation of yogurt, while the number of LAB and acid content was significantly increased compared with the control. Since PEWH displays excellent antioxidant activity, further studies are necessary to investigate the effect of added PEWH on the antioxidant capacity of yak yogurt during storage.

## 3. Material and Methods

### 3.1. Materials

Tibetan eggs were collected from local farms in Tibet (Tibet, China). Papain was obtained from the Sigma-Aldrich Chemical Co. Ltd. (Shanghai, China). All other chemicals were of analytical reagent grade.

### 3.2. The Extraction of Egg White Protein from Tibetan Eggs

Egg white proteins were extracted according to the method described by Qiu et al. [10] with some modifications. Briefly, egg white proteins were separated from the yolk and gently homogenized for 30 min with a magnetic stirrer. The obtained egg white proteins were freeze-dried and stored below −20 °C for further use.

### 3.3. The Preparation of PEWH

The hydrolysates were prepared using papain following the method described by Moreno-Fernandez et al. [14]. Briefly, the freeze-dried egg white protein was dissolved in proper distilled water and homogenized before hydrolysis. The pH value of the mixture was adjusted to about 7.0 by using either 0.01 M NaOH or 0.05 M HCl. Each of the mixtures was heated in a water bath to 37 °C before protease was added. The mixtures were incubated w stirring and then heated in a boiling water bath for 10 min to inactivate the enzymes. The hydrolysates were centrifuged at 4500 g for 15 min, after which the supernatant was collected and stored below −20 °C for further use.

### 3.4. O_2_^−^ Scavenging Activity (%) Assay

The O_2_^−^ scavenging activity of the egg white protein hydrolysates was determined by measuring the inhibition of the auto-oxidation of pyrogallol, and the procedures followed the method described by Chen et al. [31].

### 3.5. RSM

The Box-Behnken design (BBD) of RSM was used to screen the optimal conditions for producing Tibetan egg white hydrolysate. Three independent variables in this study were X_1_, X_2_, and X_3_, while the O_2_^−^ scavenging activity of the hydrolysates was used as the response variable (Y). The BBD optimized these variables and levels. The Design Expert (version: 8.0.6, Stat-Ease, Minneapolis, MN, USA,) software was employed to design the experiment and calculate the data.

The coded settings for the variables and levels in this study are shown in Table 4. A total of 17 runs containing five replicates were performed, and the following equation expressed the actual and coded values in the statistical calculation:(2)$$Y=b_{0}+\sum_{i=1}^{3}b_{i}X_{i}+\sum_{i=1}^{3}b_{ii}X_{i}^{2}+\sum_{i=1}^{2}\sum_{j=i+1}^{3}b_{ij}X_{i}X_{j}+e_{i}$$
where *Y* is the response, and *b_0_*, *b_i_*, *b_ii_*, and *b_ij_* are the model regression coefficients for intercept, linearity, square, and interaction terms, respectively, while *X_i_* and *X_j_* are the coded independent variables, *e_i_* is variate.

### 3.6. The Manufacture of Yak mIlk Yogurt Fortified with PEWH

Tibetan yak milk containing 7% fat was purchased from the Plateau Treasure Yak Dairy Co., Ltd., (Lhasa, Tibet Province, China). The manufacture of yak milk yogurt was performed according to the method outlined by Balthazar et al. [26] with modifications. Different quantities of PEWH (0% (YP0), 1% (YP1), 2% (YP2), and 3% (YP3)) were added to the yak milk. The mixtures were placed in a water bath at 90 ± 1 °C for 5 min with constant stirring. Reactivated yogurt cultures were inoculated into the mixtures when the temperature had cooled to 42 °C. The inoculated samples were maintained at 43 ± 1 °C for 0 h, 3 h, 6 h, and 12 h, after which the yogurt products were stored at 4 ± 1 °C.

### 3.7. Physicochemical Analyses and Sensory Evaluation of the Yak Milk Yogurt

The titratable acidity (TA) was determined via titration with sodium hydroxide using phenolphthalein as an indicator and was expressed as a Thorner degree (°T) value. The pH value was determined using a digital pH meter, [26] while the amino acid composition and content of samples were examined using an automatic amino acid analyzer (L-8900, HITACHI, Tokyo, Japan).

The sensory evaluation was performed using the method described by Ye et al. [30] with minor modifications. Briefly, the appearance, texture, flavor and acceptability of the yogurt were assessed by ten panelists who were experienced and knowledgeable in food science. The sensory scoring parameters were as follows: Extremely unacceptable contributed 1 point; unacceptable–barely acceptable contributed 2 to 4 points; acceptable–highly acceptable contributed 5 to 9 points; extremely acceptable contributed 10 points.

### 3.8. Viable Counts of LAB

A plate colony counting experiment was used to investigate the viable counts of LAB in different yak milk yogurt samples, according to the method described by Lin et al. [32]. The obtained results were presented in the log CFU/mL of yogurt.

### 3.9. Statistical Analysis

The results of the experiments performed in triplicate were presented as the mean ± standard deviation. Data analysis was carried out with the Duncan’s test and ANOVA using the SPSS 19.0 software (Inc., Chicago, IL, USA). The statistical significance of the difference between samples was determined at a 5% confidence level (*p* < 0.05).

## 4. Conclusions

In this study, the BBD of RSM is used to optimize the hydrolysis conditions for the preparation of antioxidant PEWH. The results indicate that the optimum conditions for O_2_^−^ scavenging activity are as follows: X_1_ of 2.51 h, X_2_ at 4.13%, and X_3_ at 4500 U/g of substrate. The predicted maximum O_2_^−^ scavenging activity approaches the experimental value, which demonstrates a good fit between the model and the actual data. Moreover, the addition of PEWH significantly affects the acidification, sensory characteristics, viable counts of LAB, and amino acid content of fermented yak milk yogurt products. Therefore, PEWH may exhibit application potential as a natural antioxidant compound in yak milk yogurt.

## Figures and Tables

**Figure 1 molecules-25-00109-f001:**
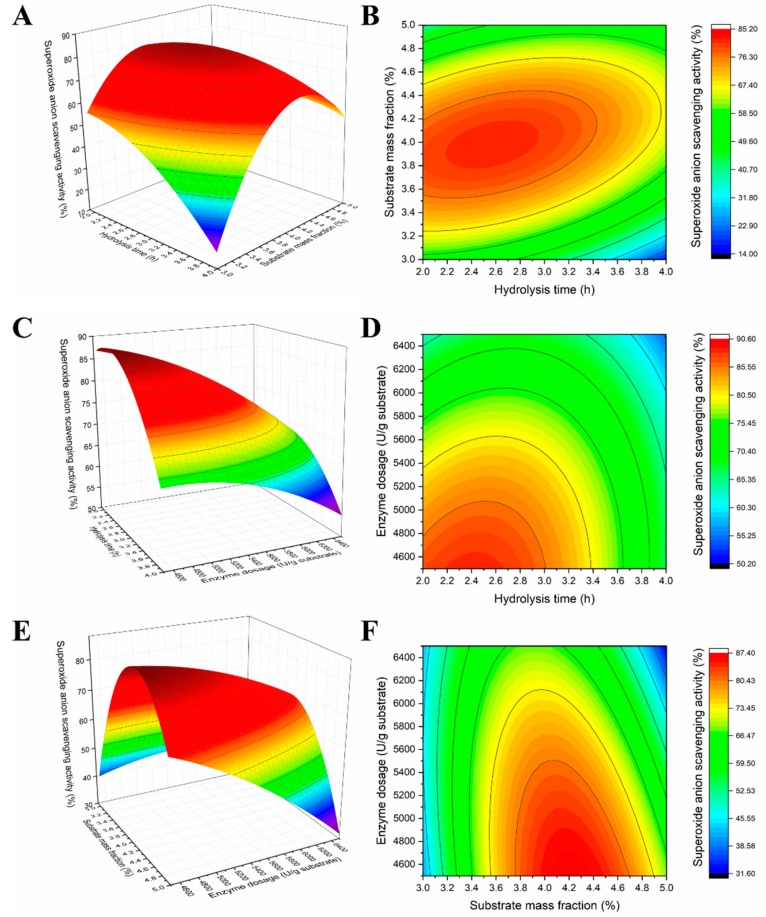
Response surface (3D) and contour plots (2D) showed that the effects of different hydrolysis parameters (X_1_: Hydrolysis time; X_2_: Substrate mass fraction; X_3_: Enzyme dosage) on the O_2_^−^ scavenging activity. ((**A**,**B**) indicated the effect of X_1_ and X2, (**C**,**D**) indicated the effect of X_1_ and X_3_, (**E**,**F**) indicated the effect of X_2_ and X_3_).

**Figure 2 molecules-25-00109-f002:**
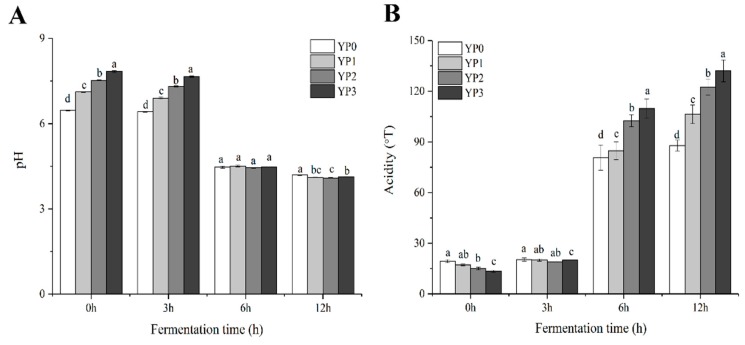
Effects of the addition of PEWH on the pH (**A**) and titratable acid (**B**) of yak milk yogurt. Different letters indicate significantly different (*p* < 0.05).

**Figure 3 molecules-25-00109-f003:**
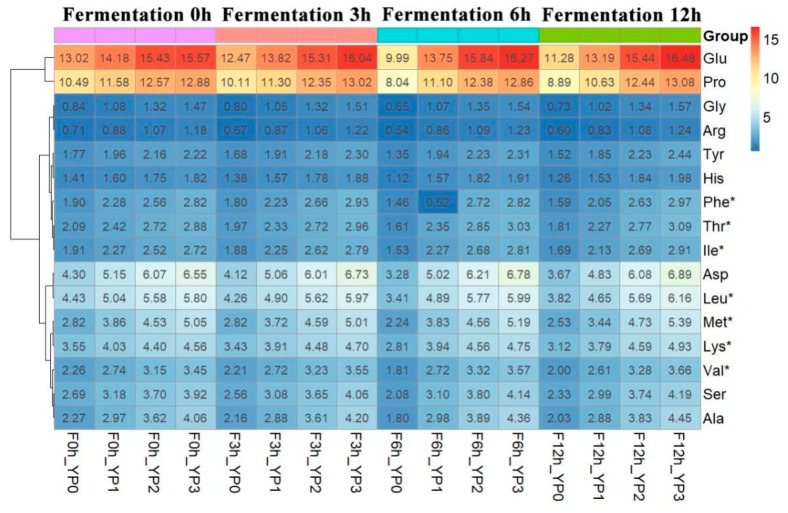
Heatmap of amino acid compositions and contents (%) during fermentation of yak milk yogurt.

**Figure 4 molecules-25-00109-f004:**
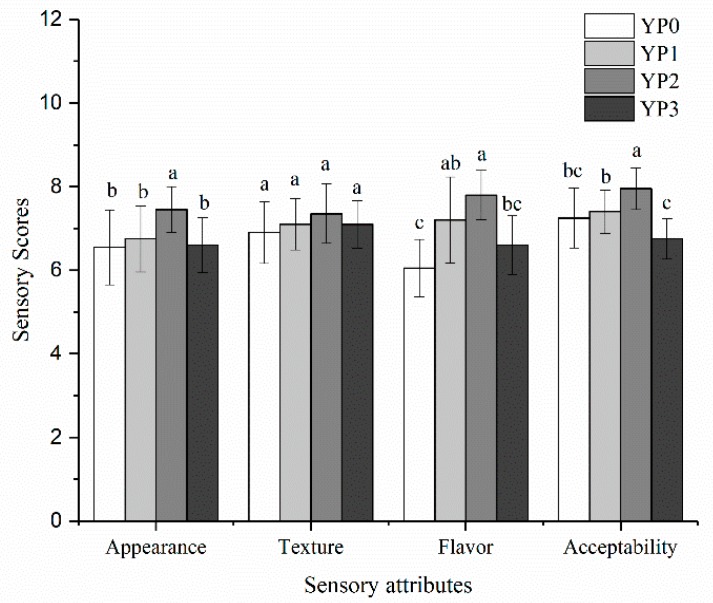
Scores of sensory indicators (D) for yak milk yogurts with different amounts of PEWH. Different letters indicate significantly different (*p* < 0.05).

**Figure 5 molecules-25-00109-f005:**
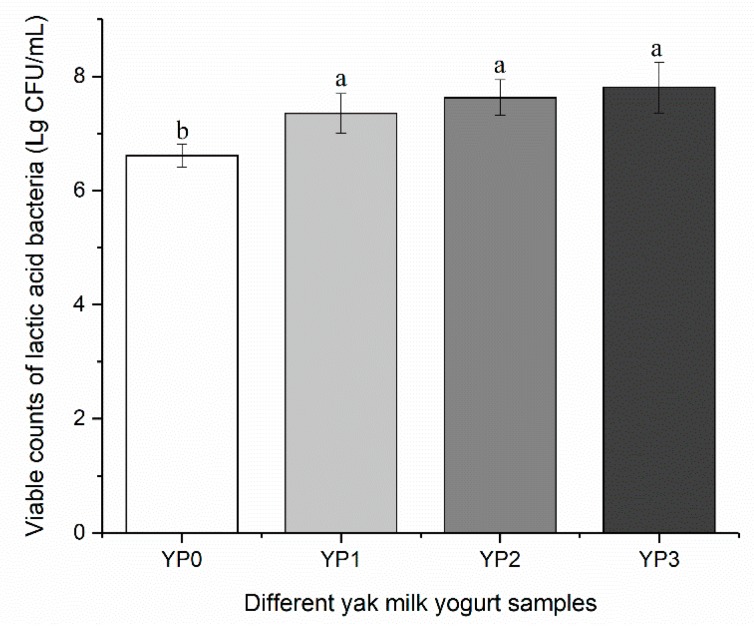
Effect of the addition of PEWH on the viable counts of lactic acid bacteria of yak milk yogurt. Different letters indicate significantly different (*p* < 0.05).

**Table 1 molecules-25-00109-t001:** BBD of RSM for optimizing hydrolysis conditions with the O_2_^−^ scavenging activity.

Run Number	Experimental Values			Y: The O_2_^−^ Scavenging Activity (%)
	X_1_: Hydrolysis Time (h)	X_2_: Substrate Mass Fraction (%)	X_3_: Enzyme Dosage (U/g Substrate)	
1	4 (1)	3 (−1)	5500 (0)	14.58
2	2 (−1)	5	5500	42.34
3	4	5 (1)	5500	47.75
4	3 (0)	4 (0)	5500	80.26
5	3	3	4500 (−1)	43.55
6	3	5	4500	71.47
7	3	4	5500	76.55
8	3	4	5500	81.56
9	2	4	4500	82.11
10	4	4	4500	66.55
11	3	3	6500 (1)	39.94
12	3	4	5500	84.71
13	4	4	6500	59.68
14	2	3	5500	58.75
15	3	5	6500	29.13
16	2	4	6500	63.70
17	3	4	5500	78.73

**Table 2 molecules-25-00109-t002:** Variance analysis of mathematical regression model.

Factor	SS	DF	MS	*F*-Value	*P*-Value (Prob > F)
Model	6803.09	9	755.9	38.64	<0.0001
X_1_	425.44	1	425.44	21.75	0.0023
X_2_	143.4	1	143.4	7.33	0.0303
X_3_	634.21	1	634.21	32.42	0.0007
X_1_X_2_	614.54	1	614.54	31.41	0.0008
X_1_X_3_	33.29	1	33.29	1.7	0.2333
X_2_X_3_	375	1	375	19.17	0.0032
X_1_^2^	323.09	1	323.09	16.51	0.0048
X_2_^2^	3980.6	1	3980.6	203.46	<0.0001
X_3_^2^	54.33	1	54.33	2.78	0.1396
Residual	136.95	7	19.56		
Lack of fit	99.41	3	33.14	3.53	0.1273
Pure error	37.55	4	9.39		
Total variation	6940.04	16			
R^2^ = 0.9803; Adj R^2^ = 0.9549; C.V.% = 7.36					

**Table 3 molecules-25-00109-t003:** Predicted and experimental values of the responses at optimum conditions.

Items	Values
**Independent variables**	
X_1_: Hydrolysis time (h)	2.51
X_2_: Substrate mass fraction (%)	4.13
X_3_: Enzyme dosage (U/g substrate)	4500
**Response values: O_2_^−^ scavenging activity**	
Predicted values	89.06%
Actual values	88.05 ± 1.2%

**Table 4 molecules-25-00109-t004:** Coded settings for the process parameters for hydrolysis, according to a BBD design of RSM.

Level	Parameter		
	X_1_: Hydrolysis Time (h)	X_2_: Substrate Mass Fraction (%)	X_3_: Enzyme Dosage (U/g Substrate)
−1	2	3	4500
0	3	4	5500
1	4	5	6500

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
