# Peer review of "Optimization of Antioxidant Hydrolysate Produced from Tibetan Egg White with Papain and Its Application in Yak Milk Yogurt"

_molecules, 2019, doi:10.3390/molecules25010109_

Round 1
Reviewer 1 Report
This manuscript reports the optimization of an antioxidant extract produced by papain protease treatment of Tibetan white egg and its application its application in yak mil yogurt. Overall, the manuscript is generally well written. The authors provide a good description of the background and rationale for their study and clearly state their experimental objectives. The materials and methods are presented clearly and with enough detail to allow repetition by others. The results are presented and discussed objectively. I recommend publication after minor corrections of English language.
Please check first sentence of abstract (L 13)
Please rephrase lines:
L 49-50
L61-64
L251-256
L283 – while?
L 250 – should be figure 3
Author Response
Dear Editors and Reviewers:
Thank you very much for your letter and valuable comments concerning our manuscript entitled “Optimization of antioxidant hydrolysate production from Tibetan egg white with papain and its application in yak milk yogurt” (Manuscript Number: molecules-660257). Those comments are all valuable and very helpful for revising and improving our paper, as well as the important guiding significance to our researches. We have studied comments carefully and have made correction which we hope meet with approval. Revised portions are marked in red in the paper. Attached please find the revised version, which we would like to submit for your kind consideration. The main corrections in the paper and the responds to the reviewer’s comments are as flowing:
Responds to the reviewer’s comments:
Response to comment: Please rephrase lines: L 49-50, L61-64, L251-256, L283-while? L250-hould be figure 3
Response: Special thanks to you for your good comments. We are very sorry for our negligence of this fault. It has been changed in the manuscript.
We appreciate for editors and reviewers’ warm work earnestly, and hope that the correction will meet with approval. Once again, thank you very much for your comments and suggestions.
Thank you and best regards.
Yours sincerely,
Liu Liu
Prof. of Shaanxi Normal University
E-mail: Prof_liuliu@126.com

Reviewer 2 Report
The paper has some problems both in analysis, spelling and grammar. In my opinion a considerable amount lab work and statistical analysis has gone into this paper but has not been transferred satisfactorily in the paper. The references used, should be more detailed and not so abrupt. For example, in the introduction a few things could be mentioned about the relationship that exists among oxidation, chronic diseases and inhibition of oxidation through egg white hydrolysates. Furthermore, there is a lack of analysis in “materials and methods”. For example, there are no details on some critical points in the description of the methodology used i.e. temperature, time and reactivated yogurt cultures which were used. Additionally, there is a limited number of references in the “results and discussion” section. In each sub-section, there is only one or two citations without actually discussing and comparing the results and sometimes repeating the same phrases with different words. For instance, in the sub-chapter 3.3 you could say that “proteins and peptides also seemed to be responsible for the time taken to reach pH 4.5 and for the difference in viability of starter bacteria in yogurts supplemented with various ingredients” instead of “the ingredients added to milk considerably affect the incubation time required to reach a pH of 4.5”, reported by Dave and Shah, because it is more specific and relevant with your paper. Also, in the sub-chapter 3.4, you focus only on your results and you didn’t focus at all on other papers. You just refer to “amino acid profiles and peptides sequences play an important role in some biological activity” and the definition of essential amino acids. I would suggest to analyze the real effect of PEPW on the amino acids of yak milk yogurt and what that means and how important it is for the human body in comparison with previous papers. Furthermore, you mention below that “the addition of PEWH to yak milk will not have a negative impact on the sensory sensation of yogurts”, but I think as a conclusion you could write that the addition of 2% PEPW in milk (YP2) has a positive effect regarding the sensory scores.
Finally, there is a lot of room for improvement in the use of the English language.
I have used the comment tools of Adobe acrobat for comments and suggestions in specific parts of the paper that should in my opinion be addressed.

Author Response
Dear Editors and Reviewers:
Thank you very much for your letter and valuable comments concerning our manuscript entitled “Optimization of antioxidant hydrolysate production from Tibetan egg white with papain and its application in yak milk yogurt” (Manuscript Number: molecules-660257). Those comments are all valuable and very helpful for revising and improving our paper, as well as the important guiding significance to our researches. We have studied comments carefully and have made correction which we hope meet with approval. Revised portions are marked in red in the paper. Attached please find the revised version, which we would like to submit for your kind consideration. The main corrections in the paper and the responds to the reviewer’s comments are as flowing:
Responds to the reviewer’s comments:
Response to comment: In Line 128: did you spread ondeMan-Rogosa Sharpe agar or Bromcresol purple agar or both of them?
Response: Special thanks to you for your good comments. We refer to method of Lin et al and modify it slightly. During the experiment, we used deMan-Rogosa Sharpe agar to cultivate LAB.
Response to comment: InLine 257. ACE inhibitory activity- why do you refer some?
Response: Thank you very much for your suggestion. The addition of bioactive peptides to Yogurt, which is a popular fermented milk product, is a convenient and feasible approach for increasing the daily intake of these peptides. The increase of PEWH to yak milk can promote the content of essential amino acids, which can play an important role in biological activity.
Response to comment: In line 258, at which fermentation point sensory scores were studied and why?In line 278, why did you check the viable counts of LAB only at the end of fermentation? is that conclusion safe?
Response: Thank you very much for your suggestion. We studied sensory scores of yak milk yogurt at fermentation endpoints (12 h). It is due to that the number of lactic acid bacteria in 12 h reached the growth logarithmic stage and had a good representative.
Response to comment:The paper has some problems both in analysis, spelling and grammar. In my opinion a considerable amount lab work and statistical analysis has gone into this paper but has not been transferred satisfactorily in the paper. The references used, should be more detailed and not so abrupt. For example, in the introduction a few things could be mentioned about the relationship that exists among oxidation, chronic diseases and inhibition of oxidation through egg white hydrolysates. Furthermore, there is a lack of analysis in “materials and methods”. For example, there are no details on some critical points in the description of the methodology used i.e. temperature, time and reactivated yogurt cultures which were used. Additionally, there is a limited number of references in the “results and discussion” section. In each sub-section, there is only one or two citations without actually discussing and comparing the results and sometimes repeating the same phrases with different words. For instance, in the sub-chapter 3.3 you could say that “proteins and peptides also seemed to be responsible for the time taken to reach pH 4.5 and for the difference in viability of starter bacteria in yogurts supplemented with various ingredients” instead of “the ingredients added to milk considerably affect the incubation time required to reach a pH of 4.5”, reported by Dave and Shah, because it is more specific and relevant with your paper. Also, in the sub-chapter 3.4, you focus only on your results and you didn’t focus at all on other papers. You just refer to “amino acid profiles and peptides sequences play an important role in some biological activity” and the definition of essential amino acids. I would suggest to analyze the real effect of PEPW on the amino acids of yak milk yogurt and what that means and how important it is for the human body in comparison with previous papers. Furthermore, you mention below that “the addition of PEWH to yak milk will not have a negative impact on the sensory sensation of yogurts”, but I think as a conclusion you could write that the addition of 2% PEPW in milk (YP2) has a positive effect regarding the sensory scores.
Finally, there is a lot of room for improvement in the use of the English language.
Response: Thank you very much for your suggestion. We are very sorry for these simple grammar mistakes due to our negligence,and we have made thorough revisions in the manuscript according to your comments. The problems in references has been revised according to your opinion, for example, the relevant references have been further explained in the introduction. The problems in the material method, such as enzymatic hydrolysis temperature and pH value, have been modified according to your suggestions. Thank you for your comments. We are very sorry for doing a bad job in the result analysis, as well as modified the line 273 in results and discussion.
We appreciate for editors and reviewers’ warm work earnestly, and hope that the correction will meet with approval. Once again, thank you very much for your comments and suggestions.
Thank you and best regards.
Yours sincerely,
Liu Liu
Prof. of Shaanxi Normal University
E-mail: Prof_liuliu@126.com
